# Risk Score to Predict Dental Caries in Adult Patients for Use in the Clinical Setting

**DOI:** 10.3390/jcm8020203

**Published:** 2019-02-07

**Authors:** Miguel de Araújo Nobre, Ana Sezinando, Inês Fernandes, Paulo Maló

**Affiliations:** 1Research and Development Department, Maló Clinic, 1600-042 Lisbon, Portugal; 2Dentistry Department, Maló Clinic, 4100-130 Porto, Portugal; asezinando@maloclinics.com; 3Dentistry Department, Maló Clinic, 1600-042 Lisbon, Portugal; ifernandes@maloclinics.com; 4Clinic, Maló Clinic, 1600-042 Lisbon, Portugal; research@maloclinics.com

**Keywords:** tooth, caries, risk, epidemiology

## Abstract

Background: There is a need for risk prediction tools in caries research. This investigation aimed to estimate and evaluate a risk score for prediction of dental caries. Materials and Methods: This case-cohort study included a random sample of 177 cases (with dental caries) and 220 controls (randomly sampled from the study population at baseline), followed for 3 years. The risk ratio (RR) for each potential predictor was estimated using a logistic regression model. The level of significance was 5%. Results: The risk model for dental caries included the predictors: “presence of bacterial plaque/calculus” (RR = 4.1), “restorations with more than 5 years” (RR = 2.3), “>8 teeth restored” (RR = 2.0), “history/active periodontitis” (RR = 1.7) and “presence of systemic condition” (RR = 1.4). The risk model discrimination (95% confidence interval) was 0.78 (0.73; 0.82) (*p* < 0.001, C-statistic). Patients were distributed into three risk groups based on the pre-analysis risk (54%): low risk (<half the pre-analysis risk; caries incidence = 6.8%), moderate risk (half-to-less than the pre-analysis risk; caries incidence = 20.4%) and high risk (≥the pre-analysis risk; caries incidence = 27%). Conclusions: The present study estimated a simple risk score for prediction of dental caries retrieved from a risk algorithm with good discrimination.

## 1. Introduction

Dental caries is the most prevalent disease worldwide. According to the global burden of oral disease report of 2005 [1], the prevalence of dental caries among adults is nearly 100% of the population in the majority of countries. 

The negative impact of dental caries on quality of life is significant both in the short and long term, with edentulism in senior patients reaching as high as 26–50% in North America and 13–78% in Europe [1]. Furthermore, regarding the global burden of oral diseases of 2010, caries are responsible collectively with periodontal disease, edentulism, oral cancer and cleft lip/palate for 18,814,000 disability-adjusted life-years (number of lost years of healthy life), corresponding to an increase by an average of 45.6% from 1990 to 2010 [2]. The negative influence of dental caries impacts far beyond overall health, affecting other important aspects of life such as social and employment opportunities [3]. 

Dental caries are dependent on lifestyle and dietary factors [4,5], with diet and smoking suggested as potential risk factors in a recent consensus [5]. Considered a multifactorial disease, several variables were proposed as potential risk factors/indicators for dental caries, including the previous or current presence of caries, number of teeth present, or plaque index [6]. The fact that lifestyle factors have important effects not only on dental caries but also on diabetes management rendered the suggestion of appropriate risk factor management procedures to be adopted in the dental setting such as the reduction of sugar consumption in order to increase the probability of a successful management of patients at risk of one or both diseases [7].

The multifactorial origin of most chronical conditions represents a challenge for clinicians to correctly diagnose and manage the risk of a specific patient for developing that condition. Risk algorithms for disease modeling assumes a central role in modern medicine as it allows the clinicians to access a tool to aid the diagnostic and decision process [8]. 

The aim of this study was to estimate and evaluate a risk score to predict dental caries for adult patients’ stratification.

## 2. Materials and Methods

The study was approved by an independent ethical committee (Ethical Committee for Health, authorization No. 005/2012). The study was conducted in full accordance with ethical principles, including the World Medical Association Declaration of Helsinki (version 2008). Written informed consent was obtained from each participant according to the aforementioned principles. 

This was a prospective case-cohort study based on the data collected from a prospective epidemiological surveillance study on oral diseases [9]. The data was collected from July 2012 to December 2015. An initial cohort of 22,009 participants was established. From these, there were 19,868 patients with natural teeth. All patients were part of a recall control maintenance program with diagnosis, prophylaxis and motivation indications including plaque control, dietary habits and fluoride daily use. Patients with presence of dental caries in the first observation were excluded from this study. A total of 5877 patients with teeth and without dental caries in the first observation were eligible for inclusion, from which, 1151 registered dental caries (attack rate of 19.6%) and 4726 registered a healthy dentition during the follow-up of the epidemiological surveillance study. A total of 397 patients were included in this study (*n* = 177 cases; *n* = 220 controls).

Examinations were performed by 22 clinicians trained and calibrated to diagnose and differentiate between sound surfaces and caries lesions both clinically and radiographically. Training and reliability assessment of dental examinations was conducted within the same day with 30 patients in each annual workshop session, resulting in 90 observations per clinician during the three workshop sessions. The overall inter-examiner reliability was estimated using a weighted average of the pairwise inter-examiner reliability estimates. The reliability of outcome assessors collecting clinical information assessed through the weighted kappa scores during the three years of follow-up were 0.88, 0.89 and 0.87, respectively [9]. One trained outcome assessor was responsible for collecting information from the records.

### 2.1. Sample Size Calculation and Sampling

The sample size calculation was performed using a software program [10]. The authors planned a study with independent cases and controls with 1 control(s) per case. Prior data indicated that the probability of exposure among controls (taking bacterial plaque as a reference) was 19% [11]. If the true odds ratio for disease in exposed subjects relative to unexposed subjects is 2, we will need to study 177 case patients and 177 control patients to be able to reject the null hypothesis that this odds ratio equals 1 with a probability (power) of 80%. The Type I error probability associated with this test of this null hypothesis is 5%. Given the nature of the study design (case-cohort study) the number of controls had to be necessarily adjusted. Considering the number of controls calculated in the previous step (*n* = 177) and the “attack rate” of 19.6% $\left[\left(\frac{Patients\;with\;caries}{Total\;patients}\right)\Leftrightarrow \left(\frac{1158}{5877}\right)\right]$, the sample size equation to adjust the number of controls in a case-cohort study design considered [12]: (1)$$N=number\;of\;controls\;\left[\frac{1}{1-atack\;rate}\right]$$
$N=177\;\left[\frac{1}{1-0.196}\right]$ ⇔ $N=177\;\left[\frac{1}{0.804}\right]$ ⇔ N=177 [1.244] ⇔ N=220 controls.

A total of 397 patients were selected. The 177 cases (with dental caries) were selected randomly from the sample of 1151 patients with dental caries. Given the nature of the study design (case-cohort design), the 220 controls were selected randomly from the global sample of 5877 patients (controls could have the presence or absence of dental caries at the end of the study follow-up given the random sample). The random sampling was performed using a random numbers generator (www.random.org).

### 2.2. Variables Definition

The dependent variable “dental caries” was defined as cavitated dentin carious lesions or the presence of a new restoration, according to WHO criteria [13]; or the lesion visible in a bitewing radiographic survey (recorded between C1 and C5 of the Mejare criteria for bitewing radiolucency scores) [14]. Baseline radiographic and clinical evaluations were performed to attest the absence of dental caries [9].

The independent variables were: age, gender (male or female), socio-economic status evaluated from the occupation of each patient according to the Goldthorpe classification (1: Higher managerial, administrative and professional occupations; 2: Intermediate occupations; 3: Routine and manual occupations) [15], systemic conditions (presence or absence); smoking (smoker, non-smoker), anti-depressant medication (presence, absence), periodontitis defined as the inflammation of the gingiva and the adjacent attachment apparatus with loss of clinical attachment and loss of adjacent supporting bone loss according to the American Academy of Periodontology [16]; active or as history of periodontitis if the patient had a controlled state of periodontitis with pocket depths </= 4 mm without ongoing clinical attachment or marginal bone loss: presence, absence), dental crowding according to Chu’s classification [17] (presence, absence), number of teeth (total number of teeth present in the mouth), bacterial plaque according to Silness and Löe index [18]/ calculus (presence or absence), number of restored teeth (recoded as: “</= 8 teeth” or “more than 8 teeth”), restorations with more than 5 years considering the follow-up time reported in the literature for composite resin restorations [19] (presence or absence) and previous experience of caries (presence or absence).

### 2.3. Statistical Analysis

#### 2.3.1. Assessment of Risk Factors

Descriptive statistics were applied for all variables (measures of central tendency and variance for continuous variables, ratios and frequencies for dichotomous variables). To retrieve the risk model, the statistics were performed according to previously described methods [20]. Univariate logistic regression was performed to all independent variables with estimation of the crude risk ratio (RR) with 95% confidence interval (CI). The independent variables significantly related to the outcome variable at the univariate analysis (*p* < 0.100) were inserted into a multivariable logistic regression model. The variables were introduced in the model without stepwise elimination, a direct estimate of the adjusted RRs (95% CI) was obtained from the model output and standard errors of the RR were adjusted through the robust variance estimator method [20]. The significance level was set at 5%.

#### 2.3.2. Development of a Risk Score

The authors developed the risk score, based on previously described statistical methods [21,22,23]. The independent predictor with lowest beta coefficient (systemic conditions: 0.306) was assigned one point of prediction score and the prediction points for the other predictors were based on the ratio of beta coefficient of each predictor to the minimum beta coefficient predictor (systemic conditions). A sum of weight points for each predictor was calculated to define the final score. In order to compare the incidence of dental caries, the patients were divided into three risk groups. The cutoff points for the three risk groups were defined based on the multiples of the pre-analysis risk: less than half the pre-analysis risk (low risk), more than half the pre-analysis risk and less than the pre-analysis risk (moderate risk), and more than the pre-analysis risk (high risk). Robust beta coefficients were calculated for the risk groups after bootstrapping based on 1000 bootstrap samples with bias-corrected accelerated 95% CI. The risk score discrimination was expressed by the C statistic (95% CI). Statistics were performed using the SPSS version 17 (Statistical Package for Social Sciences, IBM SPSS, New York, NY, USA).

## 3. Results

### 3.1. Participants

The sample of 397 patients included in the study had an average age (standard deviation) of 51.4 years (14.9 years), with a gender distribution of 226 female patients (56.9%) and 171 male patients (43.1%) (Table 1). There were 214 patients with dental caries (53.9%), and 183 patients without dental caries (46.1%). The average time of follow-up free from dental caries (standard deviation) was 17.8 months (10.0 months) for cases and 35.1 months (4.2 months) for controls. 

### 3.2. Risk Model

The variables significantly associated with dental caries and included in the risk model as predictors were: the presence of bacterial plaque/calculus (RR = 4.1), the presence of restorations with more than 5 years (RR = 2.3), the presence of >8 teeth restored (RR = 2.0), history/active periodontitis (RR = 1.7) and the presence of a systemic condition (RR = 1.4) (Table 2). All the variables included in the model fulfilled the criteria for the absence of significant multicollinearity (tolerance > 0.2, variance inflation factor < 2). The risk model provided a discrimination (95% CI) of 0.78 (0.73;0.82) (*p* < 0.001, C-statistic; Figure 1). 

### 3.3. Risk Score

The algorithm used to determine the predicted probability of dental caries according to the risk points and risk groups is illustrated in Table 3. The patients were distributed into three risk groups based on multiples of the pre-analysis risk of the data (54%): the low risk group considered the patients with less than half the pre-analysis risk; the moderate risk group considered the patients with half to less than one times the pre-analysis risk; while the high risk group considered the patients with one or more times the pre-analysis risk. The observed incidence of dental caries in the low-, moderate-, and high-risk groups was 6.8%, 20.4% and 27%, respectively. A real-life clinical situation illustrates the practical use of the risk score (Figure 2, Figure 3 and Figure 4).

## 4. Discussion

The main finding of the present study was the estimation and validation of a simple risk score for prediction of dental caries that enabled risk stratification: a score of less than 3 points to distinguish low risk, a score between 4–7 points for moderate risk and a score of 8 or more points for high risk. The final application of this research is to be used in a clinical setting to help stratify patients and provide tailored preventive actions, recall regimens and diet counseling. 

Risk scores are a useful tool to motivate patients into changing risk behaviors; while for clinicians, it enables simplification of complex statistical calculations used in multi-factorial analysis and their inclusion in clinical practice [23]. In a perspective of a patient-centered prevention/management program for dental caries, it is necessary to differentiate patients at different levels of risk that potentially require different preventive measures. Certainly, while clinical evaluations based on the diagnosis of lesions only dichotomize patients into diseased or healthy, the risk score provides two thresholds. First, patients with a score lower or equal than 3 points be positively considered as low-risk patients. Patients with a score between 4 and 7 points can be perceived as moderate risk patients. For these patients, health promotion about dietary and dental hygiene habits could be conducted; prevention measures may be implemented and closer clinical/radiographic follow-up could be proposed. Moreover, patients with a score equal or greater than 8 points can expectantly be considered as high-risk patients, for whom action is required by considering the previous risk group measures, together with fluoride prescription/topical application and a tight clinical/radiographic follow up. Undeniably, higher risk patients will have the presence of more risk factors and will be prone to carious lesions [24]. The existing clinical risk assessment tools for dental caries management such as the “Caries Management By Risk Assessment” (CAMBRA) also point to the approach, providing the ability to identify candidate patients for more intensive preventive therapy [25].

The case-cohort design [26] was chosen to conduct our study as an alternative to full cohort design when data collection and follow-up is time-consuming and expensive. The case-cohort design has the particularity of randomly selecting from the source population, regardless of their disease status. Among the advantages of the case-cohort design compared to case-control studies, the fact that risk ratios can easily be obtained directly from the cross-product of exposed and unexposed cases and controls, the control group representing a random sample of the source population, and that the control group can easily be used as a reference group to investigate multiple outcomes. However, the necessity of increasing the number of controls to compensate the reduced statistical power and the necessity of an increased statistical expertise compared to traditional case-control study designs are considered limitations of the approach [12].

In the “medical model” (where dietary and host-modified biofilm are considered), the etiology disease-driving agents, balanced against protective factors and combined with risk assessment, offer the possibility of patient-centered disease prevention and management before permanent damage is inflicted to teeth [27]. The main objective of our study was the creation of a simple risk score (using only five variables) that could be performed by clinicians at any dental appointment and promptly providing the patients’ direct risk estimation and profiling, supporting the process of decision-making in both prevention and treatment. A recent systematic review [28] appraised the evidence for the prediction of caries using four caries risk-assessment systems (Cariogram, CAMBRA, American Dental Association and American Academy of Pediatric Dentistry), focusing on prospective cohort studies or randomized controlled trials. The authors concluded that the evidence on the validity for existing systems was limited and that there was a necessity to develop valid and reliable methods for caries risk assessment. Furthermore, caries risk assessment systems such as Cariogram (including 9 factors) and CAMBRA (including 25 factors) performed at a level that did not assure that including a large number of factors was more beneficial than including only a few [28]. The five variables included in the present risk score were the presence of bacterial plaque/calculus, restorations with more than 5 years, more than 8 teeth restored, history/active periodontitis and systemic conditions.

Based on the reports of previous systematic reviews, the analysis of risk indicators for dental caries was clustered in a reduced number of variables. Ritter et al. [6] reported that the variables most frequently tested and significantly associated with dental caries were baseline caries, number of teeth and plaque index, among 92 other clinical and non-clinical variables tested. Tellez et al. [28] reported that the factors common to all caries risk assessment systems were caries experience, saliva flow, diet, general health conditions, fluoride exposure and plaque. In our analysis, all three variables referenced by Ritter et al. [6] and three variables reported by Tellez et al. [28] were tested in univariate logistic regression and two of them (bacterial plaque and systemic condition) were included in the present risk score. 

Bacterial plaque represents a pathological factor with a significant association with dental caries (odds ratio range: 2.3 to 2.55) [29,30] having led to clinical guidelines suggesting plaque removal for gains in disease prevention [31]. Considering the presence of systemic conditions as a risk factor for dental caries, its significance in the model may be explained by the influence that these conditions may affect oral health in general, disturbing the host response to the plaque biofilm by upsetting the host-microbial balance [32]. Furthermore, a previous study investigating the relation between systemic disease and caries experience registered associations between systemic conditions (hepatitis, cardiovascular disease and diabetes) and asthma with higher caries experience [33]. These types of systemic conditions were also the more prevalent in our sample, accounting for a combined 91% prevalence among the patients with dental caries. 

Restorations with more than 5 years and the number of teeth restored both represent disease indicators that potentially provide high predictive values when assessing the risk of dental caries, acting as more quantitative proxy variables for caries experience. This was previously reported in systematic reviews of indicators of risk in caries management, where a previous caries experience was considered an important predictor in community-based studies [34,35]. The restoration time in function was previously used as a risk indicator in the CAMBRA tool, where the presence of restorations with up to three years registered a significant association with an increase of 46% in the odds ratio for dental caries [30]. The association between a history/presence of periodontitis and dental caries disclosed in our analysis finds parallels in the literature: a previous study that investigated factors associated with root caries incidence in an elderly population [36] registered attachment loss as a significant predictor. Another study [37] investigated the prevalence and simultaneous occurrence of periodontitis and dental caries in a population of 5255 subjects with 30 or more years and concluded that patients with periodontal disease exhibited a 10% increase in the experience of dental caries (dental caries rate = 33%) compared to patients without periodontal disease (dental caries rate = 23%). 

The limitations of this study are related to the study design, lack of control for other potential variables of interest, the short-term follow-up and the study setting (private practice). Concerning the study design, as the most prevalent disease globally [1], dental caries imply a great difficulty to study risk factors using only incident cases in a cohort study design. Attempting to study incident cases would require an extensive time-consuming procedure, which together with the potential susceptibility to sample erosion would render the investigation virtually unfeasible. In an attempt to shorten that limitation, the case-cohort design was chosen and its statistical limitations suppressed, rendering valid estimations [12]. As a global disease with multifactorial origin, dental caries may be influenced by different risk factors in different populations, including the potential influence of environmental, socio-demographic, behavioral, microbiological, dietary/nutritional, salivary, and host risk factors [34,38]. This implies from a causal component point of view that different combinations of risk factors may produce different causal mechanisms in different populations [39]. The lack of control for other potential risk indicators such as dietary habits [4], exposure to fluoride, salivary flow and testing for the level of cariogenic bacteria [40] constitute potential limitations of this study. Furthermore, the short-term follow-up of this study may imply an underestimation of the dental caries prevalence. The study setting, being conducted in a private practice may imply caution in extrapolating the results to the community, as illustrated by the distribution of the variable socioeconomic status, with a larger distribution of patients in the first category (higher status) compared to the third category (lower status). Further longitudinal prospective studies in different populations are needed in the future to confirm the generalizability of our results and further refine the model by including other potential diagnostic tests such as the detection of lactic acid [40] in order to increase the models’ sensitivity and specificity.

## 5. Conclusions

The present study estimated and validated a simple risk score for predicting dental caries and performing appropriate risk stratification. The simple risk score may enable proper clinical interventions and elucidate the patient regarding self-perceived oral health status, with the final objective of providing health gains.

## Figures and Tables

**Figure 1 jcm-08-00203-f001:**
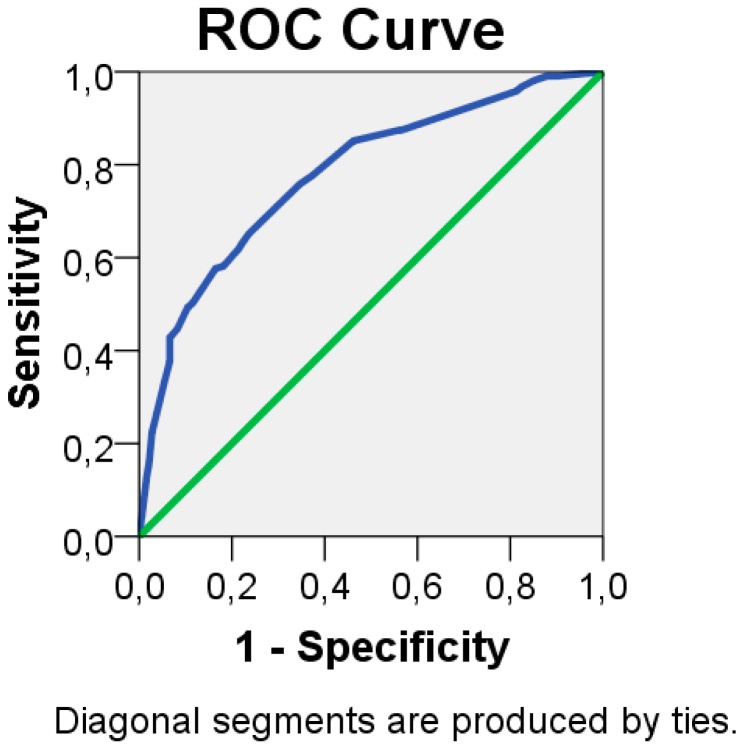
Receiver operating characteristic (ROC) curve illustrating the performance of the risk model: the model registered an area under the curve (95% confidence interval) of 0.78 (0.73;0.82) allowing good discrimination.

**Figure 2 jcm-08-00203-f002:**
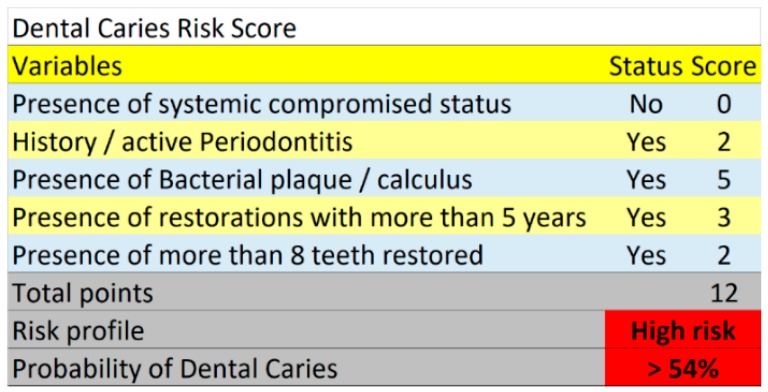
On November 2013, a 44-year-old female patient with an absence of systemic conditions, history of periodontitis, presenting bacterial plaque/calculus, restorations with more than 5 years and more than 8 teeth restored. A total of 12 points were registered in the risk score, placing the patient in the high-risk category for dental caries.

**Figure 3 jcm-08-00203-f003:**
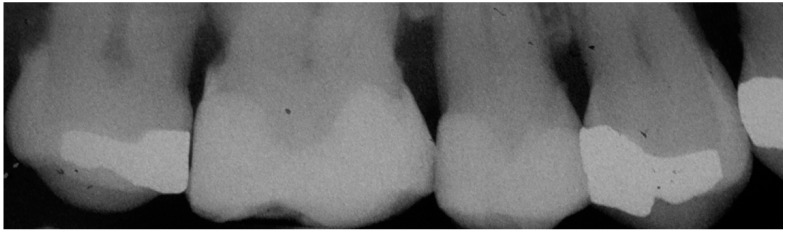
Radiographic view on November 2013 of the first sextant with tooth of interest (second premolar) with absence of dental caries. The patient missed the control appointment scheduled for 4 months.

**Figure 4 jcm-08-00203-f004:**
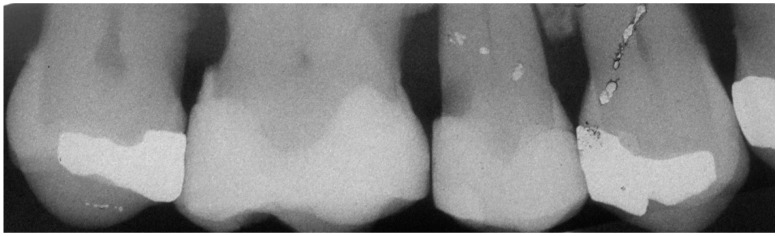
Radiographic view on November 2014 (one year after) of the first sextant with tooth of interest (second premolar) exhibiting dental caries on the distal aspect.

**Table 1 jcm-08-00203-t001:** Sample characteristics and distribution according to cases and controls.

	Total (%)	Cases (% Total)	Controls (% Total)
Number	397 (100%)	177 (44.6%)	220 (55.4%)
Age average (standard deviation)	51 (14.9)	52 (15.6)	51 (14.4)
Gender			
Female	226 (100%)	93 (41.2%)	133 (58.8%)
Male	171 (100%)	84 (49.1%)	87 (50.9%)
Systemic conditions *			
Absent	288 (100%)	120 (41.7%)	168 (58.3%)
Present	109 (100%)	57 (52.3%)	52 (47.7%)
Smoking			
Non-smoker	295 (100%)	134 (45.4%)	161 (54.6%)
Smoker	102 (100%)	43 (42.2%)	59 (57.8%)
Antidepressant medication			
Absent	371 (100%)	162 (43.7%)	209 (56.3%)
Present	26 (100%)	15 (57.7%)	11 (42.3%)
Periodontitis (history or active)			
Absent	252 (100%)	97 (38.5%)	155 (61.5%)
Present	145 (100%)	80 (55.2%)	65 (44.8%)
Dental Crowding			
Absent	288 (100%)	124 (43.1%)	164 (56.9%)
Present	109 (100%)	53 (48.6%)	56 (51.4%)
Average number of teeth present	20.6 (7.1%)	21.2 (7.0%)	20.2 (7.2%)
Bacterial Plaque/Calculus			
Absent	259 (100%)	83 (32.0%)	176 (68.0%)
Present	138 (100%)	94 (68.1%)	44 (31.9%)
Previous caries experience			
Absent	16 (100%)	5 (31.2%)	11 (68.8%)
Present	381 (100%)	172 (45.1%)	209 (54.9%)
Restoration > 5 years			
Absent	67 (100%)	17 (25.4%)	50 (74.6%)
Present	330 (100%)	160 (48.5%)	170 (51.5%)
Number of restored teeth			
</=8 teeth	226 (100%)	80 (35.4%)	146 (64.6%)
>8 teeth	171 (100%)	97 (56.7%)	74 (43.3%)
Socioeconomic status **			
1^st^ category	108(100%)	43(39.8%)	65(60.2%)
2^nd^ category	204(100%)	96(47.1%)	108(52.9%)
3^rd^ category	48(100%)	23(47.9%)	25(52.1%)

* Hepatitis (*n* = 4 patients); Cardiovascular condition (*n* = 76 patients); Thyroid condition (*n* = 15 patients); Diabetes (*n* = 12 patients); Rheumatologic 5 (*n* = 15 patients); Cancer (*n* = 6 patients); Neurological condition 11 (*n* = 1 patient); *n* = 18 patients with more than one condition. ** 1^st^ category: Higher managerial, administrative and professional occupations; 2^nd^ category: Intermediate occupations; 3^rd^ category: Routine and manual occupations.

**Table 2 jcm-08-00203-t002:** Univariable and multivariable risk ratio estimates, multivariable beta coefficients, and risk score points for the prediction of dental caries.

Variables	Univariable Risk Ratio (95% Confidence Intervals)	Univariable *p*-Value	Multivariable Risk Ratio (95% Confidence Intervals) *	Multivariable *p*-Value	Multivariable β Coefficient after Validation (Standard Error)	Risk Score Points
Age	1.0 (0.99;1.02)	0.522				
Gender						
Female	1.0 (reference)					
Male	1.38 (0.93;2.06)	0.114				
Systemic condition						
Absent	1.0 (reference)		1.0 (reference)			
Present	1.54 (0.99;2.39)	0.058	1.35 (0.83;2.22)	0.224	0.306 (0.251)	1
Smoking						
Non-smoker	1.0 (reference)					
Smoker	0.88 (0.56;1.38)	0.567				
Antidepressant medication						
Absent	1.0 (reference)					
Present	1.76 (0.79;3.93)	0.169				
Periodontitis						
Absent	1.0 (reference)		1.0 (reference)			
Present	1.97 (1.30;2.98)	0.001	1.69 (1.07;2.66)	0.024	0.523 (0.232)	2
Dental crowding						
Absent	1.0 (reference)					
Present	1.25 (0.80;1.95)	0.320				
Bacterial plaque/calculus						
Absent	1.0 (reference)		1.0 (reference)			
Present	4.53 (2.91;7.06)	<0.001	4.12 (2.59;6.55)	<0.001	1.415 (0.237)	5
Previous experience of caries						
Absent	1.0 (reference)					
Present	1.81 (0.62;5.31)	0.280				
Restorations > 5 years						
Absent	1.0 (reference)		1.0 (reference)			
Present	2.77 (1.53;5.0)	0.001	2.30 (1.21;4.37)	0.011	0.834 (0.327)	3
Number of restored teeth						
</=8 teeth	1.0 (reference)		1.0 (reference)			
>8 teeth	2.39 (1.59;3.6)	<0.001	2.03 (1.3;3.19)	0.002	0.710 (0.229)	2
Number of teeth present	1.02 (0.99;1.05)	0.198				
Socioeconomic status		0.431				
1^st^ category	1.0 (reference)					
2^nd^ category	1.34 (0.84;2.16)	0.221				
3^rd^ category	1.39 (0.7;2.76)	0.345				

* R^2^ = 0.234; Area under the curve (95% confidence interval) = 0.78 (0.73;0.82).

**Table 3 jcm-08-00203-t003:** Observed incidence of dental caries in the three risk groups.

Risk Score (Sum of Points)	Risk Group and Predicted Probability Estimated from the Risk Score	Within Group Incidence of Dental Caries	Observed Incidence of Dental Caries
0–3 points	<27%-low risk (<0.5 pre-analysis risk)	27/107 = 25.2%	27/397 = 6.8%
4–7 points	27%–54%-moderate risk (0.5 to < 1 times pre-analysis risk)	81/163 = 49.7%	81/397 = 20.4%
>/=8 points	>54%-high risk (>1 times pre-analysis risk)	107/127 = 84.3%	107/397 = 27.0%

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
