# Peer review of "Risk Score to Predict Dental Caries in Adult Patients for Use in the Clinical Setting"

_jcm, 2019, doi:10.3390/jcm8020203_

Reviewer 1 Report

RE: Risk score to predict dental caries in adult patients for use in the clinical setting.

The reviewer's concerns are as follows;

L35: What does 18 814 000 mean?

L66-67: More detail explanation for evaluating kappa scores is required. How did the authors calibarte the outcome assessment?  The X-ray films were used for agreement evaluation?

L91-93: Radiographs were taken for all teeth? What do C1-C5 mean? The root surface caries were also evaluated?

L99-102: Please explain more about periodontitis classification. All remaining teeth were examined for periodontitis evaluation? Pocket depth was measured? How is active periodontitis? How did authors diagnose those with crowding? Is there any specific index to evaluate crowding? Why did authors set cut-off level as 8 teeth? Why did authors set cut-off as 5 years?

Overall, the fluoride usage, dietary habit, regular dental visit for tooth cleaning, salivary flow rate are also affect caries incidence strongly.  There is no information. Why?

It is now easy to evaluate the level of cariogenic bacteria using commercial test kit instead of rough (visual?) estimate for plaque (presence/absence). 

Author Response

The authors thank the Reviewer for taking the time to review our manuscript.

Please find below the response to the review.

1. L35: What does 18 814 000 mean?

Response: The authors thank the Reviewer's query. The number means the total number of disability adjusted life years (DALY) that oral diseases are responsible for. It represents a metric of the World Health Organization that is interpreted the following way: One DALY can be thought of as one lost year of "healthy" life. The authors included a small explanation on the manuscript to make it clear for the reader.

Changes: Introduction section, line 35

 2. L66-67: More detail explanation for evaluating kappa scores is required. How did the authors calibarte the outcome assessment?  The X-ray films were used for agreement evaluation?

Response: The authors thank the Reviewer's query. More detail in the explanation for evaluating kappa scores was introduced in the manuscript as requested:

Examinations were performed by 22 clinicians that were trained and calibrated to diagnose and differentiate between sound surfaces and caries lesions both clinically and radiographically. Training and reliability assessment of dental examinations was conducted within the same day with 30 patients in each annual workshop session, resulting in 90 observations per clinician during the three workshop sessions. The overall inter-examiner reliability was estimated using a weighted average of the pairwise inter-examiner reliability estimates.

Changes: Materials and Methods section, lines 68-75.

3. L91-93: Radiographs were taken for all teeth? What do C1-C5 mean? The root surface caries were also evaluated?

 Response: The authors thank the Reviewer’s query. Bitewing radiographs are taken for all posterior segments teeth (that represent difficult areas of clinical screening). The Mejare criteria for bitewing radiolucency scores is a classification that varies between C0 and C5:

C0 No radiolucency evident (not recorded);

C1 Radiolucency is evident within the outer half of enamel;

C2 Radiolucency extends to the inner half of enamel and may reach the Dentin Enamel Junction (DEJ);

C3 Radiolucency extends just beyond the DEJ;

C4 Radiolucency is evident within the outer third of dentine;

C5 Radiolucency extends to the inner two thirds of dentine and may reach the pulp;

In addition, the clinical assessment was also performed in all evaluations, including root surface caries that is covered in the World Health Organization (WHO) criteria already present in the text. The authors included the information of the Mejare criteria in the text for clarity.

Changes: Materials and Methods section, Lines 100,101.

4. L99-102: Please explain more about periodontitis classification. All remaining teeth were examined for periodontitis evaluation? Pocket depth was measured? How is active periodontitis? How did authors diagnose those with crowding? Is there any specific index to evaluate crowding? Why did authors set cut-off level as 8 teeth? Why did authors set cut-off as 5 years?

Response: The authors thank the Reviewer’s queries.

Periodontitis classification: Periodontitis was defined as inflammation of the gingiva and the adjacent attachment apparatus with loss of clinical attachment and loss of adjacent supporting bone loss and according to the American Academy of Periodontology (2000). All teeth were examined for periodontitis evaluation using the definition above, if the patient had a controlled state of periodontitis (pocket depths

Dental crowding was classified considering the Chu’s classification (mesio-distal and bucco-lingual classification varying between 0 and 4) (Gurel et al. 2008). The information and reference were introduced in the manuscript to clarity.

Cut-off level at 8 teeth: The decision to set the cut-off level at 8 teeth was statistical, given the variability of the number of restored teeth in the sample with 8 teeth representing the median that allowed to split the sample in two similar groups.

Cut-off level at 5 years: The cut-off at 5 years was performed considering that according to a Meta-analysis (Opdam et al. 2014) the follow-up time for composite resin restorations in the literature rarely exceeds 5 years. The information and reference were introduced in the manuscript for clarity.

Changes: Materials and Methods section, line 107-116; References section.

5. Overall, the fluoride usage, dietary habit, regular dental visit for tooth cleaning, salivary flow rate are also affect caries incidence strongly.  There is no information. Why? 

It is now easy to evaluate the level of cariogenic bacteria using commercial test kit instead of rough (visual?) estimate for plaque (presence/absence). 

Response: The authors thank the Reviewers queries. The study was performed in a clinical setting where all patients had regular dental visit for tooth cleaning and use fluoride on a daily basis (the information was not introduced in the manuscript and should have been introduced). The authors introduced the information in the manuscript. Dietary habits were always instructed but no survey was given to the patients to record their dietary habits as well as no salivary flow rate was measured and therefore these were already considered as study limitations and acknowledged in the Discussion section. It is true that the use of commercial test kits to evaluate cariogenic bacteria is now easy. However, it represents an extra investment for a General Practitioner Dentist and/or the patient that will represent an increase in cost. This risk score was designed to be performed in a simple and practical chair-side intervention without the need of any different materials rather than the ones already available in a regular screening visit to the dental clinic, making it possible to use irrespective of the location of the dental clinic in a developed or under-development country. Other risk scores like the CAMBRA use a clinical approach independent of a commercial kit, making it’s use potentially more general for the population. Nevertheless, the information that no commercial kit to evaluate the level of cariogenic bacteria was used was acknowledged as a study limitation in the Discussion section in addition to the previous insertion in the directions of future research at the end of the Discussion section.  

Changes: Materials and Methods section, lines 61-63; Discussion section, lines 290,291.

Reviewer 2 Report

The authors have attempted to evaluate a risk score, using five variables, for prediction of dental caries in the manuscript titled “Risk score to predict dental caries in adult patients for use in the clinical setting”. Since dental caries is the most prevalent disease encountered by dentists, I think such risk score predictors are highly required in the dental setting.

Following are my concerns if they can be addresses;

(1)  Page 2, line 53 ………change “and” to “an”

(2)  Page 2, line 71 …………rephrase this sentence

(3)  Was hypertension, and high cholesterol taken into consideration for systemic conditions? What were the most common systemic conditions patients in the study suffered from?

(4)  Why was antidepressant medication considered as a predictor? Were any other group of medications also considered as variables for caries prediction?

Author Response

The authors thank the Reviewer for taking the time to review our manuscript.

Please find below the response to the review.

 The authors have attempted to evaluate a risk score, using five variables, for prediction of dental caries in the manuscript titled “Risk score to predict dental caries in adult patients for use in the clinical setting”. Since dental caries is the most prevalent disease encountered by dentists, I think such risk score predictors are highly required in the dental setting.

 Response: The authors thank the Reviewer.

Following are my concerns if they can be addresses;

(1)  Page 2, line 53 ………change “and” to “an”

Response: The authors thank the Reviewer’s suggestion. The change was performed as indicated. 

Changes: Line 54.

(2)  Page 2, line 71 …………rephrase this sentence

 Response: The authors thank the Reviewer’s indication. The sentence was revised as requested for simplicity, by deleting the redundant information about the statistical program and leaving only the scientific reference.

Changes: Line 78.

(3)  Was hypertension, and high cholesterol taken into consideration for systemic conditions? What were the most common systemic conditions patients in the study suffered from?

Response: The authors thank the Reviewer’s query. Both conditions were considered for systemic conditions. The most common systemic conditions present in the sample were: Hepatitis (n=4 patients); Cardiovascular condition (n=76 patients); Thyroid condition (n=15 patients); Diabetes (n=12 patients); Rheumatologic condition (n=15 patients); Cancer (n=6 patients); Neurologic condition (n=1 patient); n=18 patients with more than one condition. The information was inserted in Table 1.

Changes: Table 1, line 151-153.

(4)  Why was antidepressant medication considered as a predictor? Were any other group of medications also considered as variables for caries prediction?

Response: The authors thank the Reviewer’s query. Anti-depressant medication was tested as a predictor because this type of medication has the potential of provoking reduced salivary flow rate and therefore increase the probability of caries. The remaining groups were treated in the variable “systemic condition”.Changes: None.

Round  2

Reviewer 1 Report

The manuscript was improved.